# Thermal Stability and Dynamic Mechanical Properties of Poly(*ε*-caprolactone)/Chitosan Composite Membranes

**DOI:** 10.3390/ma14195538

**Published:** 2021-09-24

**Authors:** Yanbo Zhang, Yaqi Wu, Ming Yang, Gang Zhang, Haiyan Ju

**Affiliations:** School of Chemistry and Chemical Engineering, Hubei Key Laboratory of Biomass Fibers and Eco-Dyeing and Finishing, Wuhan Textile University, Wuhan 430073, China; yanboz@163.com (Y.Z.); wuyaqi07@163.com (Y.W.); yangming491@163.com (M.Y.); prayerzg2019@163.com (G.Z.)

**Keywords:** poly(*ε*-caprolactone), chitosan, composite membrane, thermal stability, dynamic mechanical analysis

## Abstract

Poly (*ε*-caprolactone) (PCL) and chitosan (CS) are widely used as biodegradable and biocompatible polymers with desirable properties for tissue engineering applications. Composite membranes (CS–PCL) with various blend ratios (CS:PCL, *w*/*w*) of 0:100, 5:95, 10:90, 15:85, 20:80, and 100:0 were successfully prepared by lyophilization. The thermal stabilities of the CS–PCL membranes were systematically characterized by thermogravimetric analysis (TG), dynamic thermogravimetry (DTG), and differential scanning calorimetry (DSC). It was shown that the blend ratio of PCL and CS had a significant effect on the thermal stability, hydrophilicity, and dynamic mechanical viscoelasticity of the CS–PCL membranes. All the samples in the experimental range exhibited high elasticity at low temperature and high viscosity at high temperatures by dynamic mechanical thermal analysis (DMTA). The performances of the CS–PCL membranes were at optimum levels when the blend ratio (*w*/*w*) was 10:90. The glass transition temperature of the CS–PCL membranes increased from 64.8 °C to 76.6 °C compared to that of the pure PCL, and the initial thermal decomposition temperature reached 86.7 °C. The crystallinity and porosity went up to 29.97% and 85.61%, respectively, while the tensile strength and elongation at the breakage were 20.036 MPa and 198.72%, respectively. Therefore, the 10:90 (*w*/*w*) blend ratio of CS/PCL is recommended to prepare CS–PCL membranes for tissue engineering applications.

## 1. Introduction

With the increasing requirements of biomedical materials, the fabrication of polymer composite membranes has recently attracted wide attention and achieved interesting and promising results in various research works. Chitosan (CS), a polysaccharide obtained from *N*-deacetylation chitin, has been widely used in the fields of daily chemicals, food, and agriculture, especially in tissue engineering scaffolds because of its excellent biodegradability, biocompatibility, and antibacterial capability [1,2,3,4]. CS is much better than the other absorbable membrane materials, and is an excellent candidate for the formation of membranes, microspheres, and fibers [5,6]. Moreover, CS exhibits a positive charge and has a similar structure to glycosaminoglycans, which provides a suitable environment for cells to efficiently accomplish their biological functions, such as promoting drug absorption [7], accelerating wound healing, and inducing bone tissue regeneration [8,9,10,11]. However, due to the difficult formation of intramolecular hydrogen bonds and a mass of cyclic structures in the molecular structure, the kinematic resistance of CS molecules exacerbates and leads to low mechanical strength and poor extensibility of single CS membranes and severely limits its application in the biomedical field [12,13]. Fortunately, CS has numerous reactive groups such as hydroxyls, acetamides, and amines, which can interact with many organic polymers to enhance the CS membrane and improve its properties for use in the tissue engineering scaffold field [14,15]. 

As a semi-crystalline aliphatic polyester, poly (*ε*-caprolactone) (PCL), with complete biodegradability, drug permeability, and biocompatibility, has been approved by the FDA (USA) and extensively used as a biomedical material [16,17]. Moreover, due to the repeat units of non-polar methylene and polar ester groups in the molecule, PCL exhibits excellent flexibility and plasticity, making it easy to transform biomedical materials into various shapes [18]. Nevertheless, there is a large obstacle for PCL applications because of its relatively low thermal stability, hydrophilicity, and cell adhesion [19]. In recent studies, it has been reported that PCL could be blended with various polymers in order to improve its thermostability, stress crack resistance, hydrophilicity, and cell adhesion [20,21,22,23]. The developed composite CS–PCL scaffolds showed a faster degradation rate, more hydrophilicity, and higher thermostability, which could make them a good candidate for biomedical applications [24,25]. Numerous studies have shown that CS–PCL membranes are suitable and promising candidates as vascular grafts, wound dressing for the controlled release of drugs, and carriers of encapsulated enoxaparin [26,27,28,29]. Furthermore, recent efforts have contributed to mediating the cellular osteogenic growth peptide gene by combining amphiphilic CS, PCL, and bioglass, and gene transfection efficiency was dramatically enhanced in the experimental conditions [30]. 

In our previous research, CS–PCL membranes with various blend ratios were prepared in glacial acetic acid solution by lyophilization, and the structural characteristics and micromorphologies were investigated by a variety of detection methods [31]. The results obtained showed good compatibility between PCL and CS. The CS–PCL membranes were blended steadily by strong hydrogen bonds and new ester bonds. Moreover, the results initially identified that the 10:90 (*w*/*w*) blend ratio of CS to PCL exhibited an optimal performance level in terms of micromorphology and composite structure. 

As we know, the good thermal stability of a CS–PCL membrane plays an important role in its processing and applications. However, there are few systematic reports about the thermal stability of CS–PCL membranes. Therefore, on the basis of the structural studies, the objectives of this study are to investigate the influence of the blend ratios on the thermal stability and dynamic mechanical properties of CS–PCL membranes, as well as to provide a significant theoretical basis for their further applications in the tissue engineering field.

## 2. Experimental Procedure

### 2.1. Materials

PCL pellets (average Mn 80,000) and glacial acetic acid were all purchased from Sigma-Aldrich (St. Louis, MI, USA). Chitosan flakes (MW 190−375 KDa, 75–85% deacetylation degree, CAS number 9012-76-4) were supplied by Sigma-Aldrich (St. Louis, MI, USA). The other reagents used for the preparation of the CS–PCL membranes were of analytical grade. 

### 2.2. Preparation of CS–PCL Membranes

PCL pellets were dissolved in a 200 mL glacial acetic acid solution (8%, *w*/*w*). Then, CS flakes were added into the six prepared PCL solutions with different ratios of CS and PCL (0:100, 5:95, 10:90, 15:85, 20:80, and 100:0, *w*/*w*) and magnetically stirred for 8 h. All of the solutions mentioned were poured into glass Petri dishes and moved to the freezer overnight for solidifying. Later, the solidified solutions were transferred into a freeze-drying vessel which had already been set to −45 °C and freeze dried for 48 h to remove the solvent. After drying, the membranes were immersed into 0.5 M NaOH solutions in order to neutralize the remaining acetic acid and washed using distilled water several times. Finally, the CS–PCL membranes were air-dried and then kept in a desiccator for use.

### 2.3. Characteristics of CS–PCL Membranes 

The mechanical properties of the samples were measured under a cross-head speed of 100 mm/min by a universal testing instrument (GT-AI-7000S, Hi-Tech, Taiwan) with 150 N load cell. The CS–PCL membranes were cut into dumbbell-shaped samples by the supporting mold of a testing instrument with regular specimens of approximately 20 mm × 4 mm × 0.8 mm. All the samples were tested in quintuplicate. The averages were taken as the test results and data are presented as mean ± SD. The mean pore size and the mean porosity of the samples were characterized by a low temperature nitrogen adsorption/desorption tester (ASAP 2020, Michel, America), which was equipped with the Statistical Package for the Social Sciences program. The differences among the samples were evaluated by the Shapiro–Wilk test followed by one-way analysis of variance with *p* value < 0.05. The hydrophilicity of the samples was evaluated by using the sessile drop method with a video contact angle measurement system instrument (OCAH200, Dataphysics, Germany) at room temperature, and three samples were employed for each test. The standard deviation and the average values were calculated for each sample. The dynamic mechanical thermal properties of the CS–PCL membranes were separately examined at 1.0 and 5.0 Hz by using a dynamic mechanical thermal analyzer (DMTA242C, NETZSCH, Germany), and the experimental temperature was gradually raised from 20 to 60 °C at a rate of 0.5 °C/min. The storage modulus (E′), loss modulus (E″), and loss factor (tan *δ* = E″/E′) were recorded by a DMTA system. The thermostability of the conditioned CS–PCL membranes was tested by differential scanning calorimetry (DSC-200PC PHOX, NETZSCH, Germany) from 20 °C to 120 °C at a rate of 5 °C/min, respectively. Nitrogen was used as the purge gas at a flow rate of 50 mL/min. TG analysis was performed by a thermal gravimetric analyzer (TG-209F1, NETZSCH, Germany) from 30 °C to 800 °C at a scanning rate of 25 °C/min under a nitrogen atmosphere. The rate of mass change (*dm*/*dt*) versus temperature from the TG graphs was plotted as the derivative thermogravimetry (DTG) curve.

## 3. Results and Discussion

### 3.1. Mechanical Property

Table 1 shows the mechanical properties of CS–PCL membranes with different blend ratios. The pure CS membrane showed low mechanical strength under tensile force, which is due to the hydrogen bonds and a wide variety of cyclic structures in the molecular structure resulting in kinematic resistance. As expected, the pure PCL membrane exhibited excellent mechanical behaviors such as high tensile strength, maximum elongation, and elongation at break. It was reported that the tensile strength of human cancellous bone ranges from 4 to 12 MPa [32]. As shown in Table 1, the tensile strengths of all the CS–PCL membranes were over 12 MPa. This suggests that CS–PCL composite scaffolds are in very good agreement with the possibility to provide support for new tissues. With decreasing CS content in the composite membranes, the mechanical properties of the CS–PCL membranes enhanced significantly. When the blend ratio of CS to PCL reached 10:90, the tensile strength and elongation at break of the composite membranes were raised to approximately 20.036 MPa and 198.72%, respectively, which is a significant improvement over the reported poly (*ε*-caprolactone)-based scaffolds for human meniscal tissue [33]. However, the performance improvement of the CS–PCL membranes was not obvious when the CS content further decreased. Therefore, it was shown that the CS content in the CS–PCL composite membrane should not be too low to give a greater influence on the mechanical properties.

### 3.2. Porosity and Pore Features

The interconnected pores in CS–PCL membranes may be conducive to cell adhesion and tissue regeneration. It has been reported that a minimum of 76–81% of overall porosity for an ideal scaffold would easily allow cells to penetrate and form tissues [34]. As shown in Table 2, the mean porosity and the specific pore features of the CS–PCL membranes fluctuated as the blend ratios changed distinctly. The lower the CS content, the larger the pore size. Pure CS membranes presented lower average porosity, pore specific surface area, and pore size than those of pure PCL, which may result from the close aggregation between the CS molecular chains caused by a large number of intramolecular hydrogen bonds [35]. As a complementary therapy, PCL with a flexible microporous structure was blended with pure CS and finally overcame the lack of pores [36]. When the blend ratio was 10:90, the CS–PCL membranes exhibited excellent cumulative adsorption in the specific surface area, and the cumulative adsorption value reached 4.6281 cm^3^/g. Moreover, the cumulative desorption of the mean pore volumes and the porosity of the composite membranes with the blend ratio of 10:90 were 0.18399 cm^3^/g and 85.61%, respectively, which is consistent with the requirements of ideal biomedical scaffolds [34].

However, continuous decreases in CS content had little effect on the improvement of the composite membrane pores. Therefore, the pore size and porosity of CS–PCL membranes with the blend ratio of 10:90 were improved significantly, which would be favorable to the cell adhesion and growth in the process of repairing tissue injury.

### 3.3. Contact Angle

The hydrophilicity of the CS–PCL membranes with different blend ratios was determined by the water contact angle measurements. As shown in Figure 1, the contact angle values of the pure CS membrane and pure PCL membrane were 127.2° ± 2.3° and 87.6° ± 1.5°, respectively, which indicates that CS is a hydrophilic polymer, while PCL is a hydrophobic polymer. This phenomenon is mainly related to the molecular structures of the two polymers. A long methylene hydrocarbyl structure in the PCL molecule results in higher hydrophobicity, while a large amount of hydrophilic functional groups such as amino, hydroxyl, and ethoxyl groups in the CS molecule contribute to its higher hydrophilicity. Hence, the hydrophilicity of the CS–PCL membranes was enhanced with the addition of CS content. When the blend ratio of CS to PCL was up to 10:90, the contact angle value of the CS–PCL membranes decreased to 95.7° ± 3.7°, which implies a moderate hydrophilicity for cell attachment [37]. With the continuous increase of CS content in the CS–PCL membranes, the contact angles decreased very slowly, and improvements in the hydrophilicity of the CS–PCL membranes were negligible.

### 3.4. Dynamic Mechanical Analysis

Dynamic mechanical analysis is a popular technique for detecting polymer transitions. The blend ratios of composite membranes may have a great influence on the state of dispersion in blends and on the thermal properties of the composite membranes [38]. The mechanical properties and temperature transitions of the CS–PCL membranes were evaluated by a dynamic mechanical analyzer in temperature sweeps at constant frequency, (a) 1 Hz and (b) 5 Hz, respectively, and deformation though the linear viscoelastic range. The temperature dependences of the storage modulus (E′) and loss factor (tan *δ*) of the CS–PCL membranes with different blend ratios are presented in Figure 2. With increasing temperature, pure PCL showed storage modulus dependence as a typical viscoelastic polymer, which means it went through a glass transition from a glassy to a rubbery state. Pure CS had significantly different behavior than pure PCL; the storage modulus of the pure CS gradually increased to the high peak and finally decreased to appear as a parabola curve (Figure 2). Moreover, when the temperature went up, the tan *δ* of the pure CS presented three stages in sequence: a rapid increase at first, a moderate level, and then a sharp rise in the end, which correspondingly indicated a high elasticity at low temperature, an appropriate viscoelasticity, and finally a distinct viscosity when the temperature rose high enough. Therefore, from the obtained curves, it could be concluded that the blending of PCL and CS was not only a simple superposition of the modulus but also a compatible interaction between CS and PCL molecules, and the viscoelasticity of the composite membranes was extremely dependent on the CS content. 

In the tested temperature range, when the blend ratio of CS and PCL was 10:90, the storage modulus remained at relatively high values and the tan *δ* peaks appeared reasonably low. As shown in Figure 2a, the melting transition temperature of the CS–PCL membranes with different blend ratios was approximately 56.7 °C, while that of pure PCL was only 53.1 °C. When the constant frequency was 5 Hz, for all of the tested membranes, the loss modulus and tan *δ* (Figure 2b) temperature dependences were found to be correspondingly similar with those (Figure 2a) where the constant frequency was 1 Hz, and slightly different in the melting transition temperatures, which were 52.9 °C (pure PCL) and 56.4 °C (CS–PCL membranes), respectively. All the above results show that the CS–PCL membranes, especially with the blend ratio of 10:90, retained excellent viscoelasticity and mechanical strength when the temperature was up to 56 °C, which is consistent with the results reported in the literature [39]. 

### 3.5. Thermal Properties and Thermal Stability

Polymer thermostability usually plays a great role in processing, molding, and applications. In this study, the thermal properties of CS–PCL membranes with different blend ratios were investigated by differential scanning calorimetry (DSC) and thermogravimetric analysis (TGA) together. The DSC thermograms (Figure 3) of prepared membranes were characterized by endothermic peaks associated with the helix–coil transition, and the estimated transition temperature (*T_m_*) and enthalpy (Δ*H_m_*) of the tested samples are accordingly summarized in Table 3. Apparently, the pure CS exhibited a typical steamed endothermic peak, and the pure PCL presented a sharp and narrow characteristic endothermic peak in Figure 3. The miscibility of a composite membrane with multiple components in the amorphous state can be inspected by detecting its sole transition temperature [40]. Namely, the immiscibility of both polymers should be demonstrated by the reservation of the individual *T_m_* values of the composite components. The *T_m_* values (Table 3) of the pure CS and PCL were 86.3 °C and 64.8 °C, respectively. 

As shown in Figure 3, all the composite membranes exhibit a single *T_m_* value in the whole composition range situated between the *T_m_* values of the CS and pure PCL, which is a clear indication of the miscibility of the two polymers. Moreover, the location of the blend *T_m_* appears to be proportional to the blend ratios. This is an effective proof that CS is practically beneficial for the thermal stability improvement of the composite membranes. When the blend ratio of CS to PCL was 10:90, the *T_m_* value of the CS–PCL membranes reached 77.6 °C, which is very close to that of the 15:85 blend ratio (77.9 °C). On the other hand, the transition enthalpy associated with the endothermic peak is related to the relative interaction between polymers, and the enthalpy value reflects the renaturation strength of the composite membranes [41]. 

The above results indicate that the transition enthalpy of the CS–PCL membranes rises with increasing of CS content, which proves a good compatibility between PCL and CS in the composite membranes. The Δ*H_m_* values of the CS–PCL membranes with the blend ratios of 10:90, 15:85, and 20:80 reached 79.65 J/g, 80.13 J/g, and 81.97 J/g, respectively, which were not improved obviously with increasing CS content. This is similar to the results of the *T_m_* values. It suggests that excessive CS does not significantly contribute to the thermal stability of CS–PCL membranes. 

TG analysis was performed to detect the thermal degradation of pure PCL, pure CS, and CS–PCL membranes with different blend ratios. As shown in Figure 4a, the pure CS exhibited a gradient weight loss thermogram, while the pure PCL displayed a single stage thermal degradation. The results are closely related to both of the polymer molecular structures. The weight loss of chitosan decomposition was recorded in two stages. The first one started at 52 °C with a weight loss of 13.2%, which is assigned to the loss of water. The second stage started at 217 °C and reached a maximum at 290 °C with a weight loss of 49.8%, which corresponds to the glycosidic bond decomposition of chitosan and a further pyrolysis of polysaccharides. The weight loss of PCL was around 86.3% in the region of 372 °C to 456 °C, associated with hydrophilic ester group breakage and long-chain hydrocarbyl rupture, which is comparable to the reported literature [42]. 

The weight loss behavior of the CS–PCL membranes with different blend ratios was depicted in Figure 4c. The CS–PCL membranes presented a single-stage thermal degradation which was similar to that of the PCL. In comparison with the pure CS, the decomposition temperatures of the CS–PCL membranes were increased substantially. This means there was a great improvement in the thermal stability of the composite membranes, which could be attributed to the interaction between the CS molecules and PCL ester groups through hydrogen bonding. However, with the further increase of CS content, both the initial and final thermal decomposition temperatures of the CS–PCL membranes gradually shifted to a lower temperature. In particular, the thermal degradation of the CS–PCL membrane with a blend ratio of 20:80 started at around 200 °C and decomposed at 415 °C. Among the test range, the optimal thermal stability of the composite membranes belongs to that of the 10:90 blend ratio of CS to PCL, approximately 89% of which was degraded between 297 °C and 439 °C.

The derivative thermo-gravimetric (DTG) analysis of the pure PCL, pure CS, and CS–PCL membranes with different blend ratios is shown in Figure 4b,d. Each temperature peak in the DTG thermograms indicates the particular temperature (*T_max_*) which corresponds to the maximum degradation rate of the tested membranes [43]. The pure CS exhibits three peaks at 69 °C, 192 °C, and 287 °C of *T_max_*, while the pure PCL only shows a single peak at 417 °C. It is noteworthy that all the tested CS–PCL membranes have good thermal stability and present a single temperature peak (Figure 4d) similar to that of the pure PCL. Moreover, the *T_max_* of the CS–PCL membrane with the blend ratio of 10:90 is 416 °C, which is the highest thermal decomposition temperature among the tested composite membranes. When the blend ratio is 20:80, the *T_max_* of the CS–PCL membrane decreases to 409 °C. This means that excessive chitosan is detrimental to the thermal stability of the CS–PCL membranes, which is consistent with the results of the dynamic mechanical thermal analysis.

## 4. Conclusions

CS–PCL membranes with different blend ratios (*w*/*w*) of 0:100, 5:95, 10:90, 15:85, 20:80, and 100:0 were successfully fabricated by lyophilization. All the samples in the experimental range exhibited high elasticity at low temperature and high viscosity at high temperature. The blend ratio of PCL and CS had a distinct influence on the properties of the PCL/CS composite membranes, and especially high CS content was unfavorable to the thermal stability. Among them, the CS–PCL membranes with the blend ratio of 10:90 (*w*/*w*) showed optimal thermal stability, hydrophilicity, and dynamic mechanical viscoelasticity. The glass transition temperature of the CS–PCL membranes increased by 11.8 °C, and the initial thermal decomposition temperature went up to 86.7 °C. Additionally, the tensile strength and elongation at break reached 20.036 MPa and 198.72%, respectively. The crystallinity and porosity reached 29.97% and 85.61%, respectively. As our previous research results described in the preface, the CS–PCL membranes (10:90, *w*/*w*) were not only remarkable in terms of their micromorphology and composite structure, but also due to their excellent thermal stability. Therefore, the blend ratio of 10:90 (*w*/*w*) is experimentally recommended to prepare CS–PCL membranes for tissue engineering applications.

## Figures and Tables

**Figure 1 materials-14-05538-f001:**
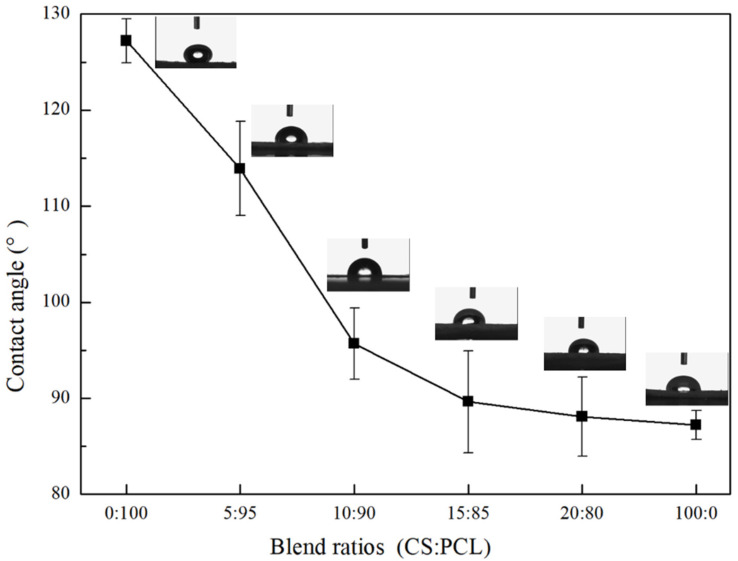
Contact angles of the tested membranes with different blend ratios.

**Figure 2 materials-14-05538-f002:**
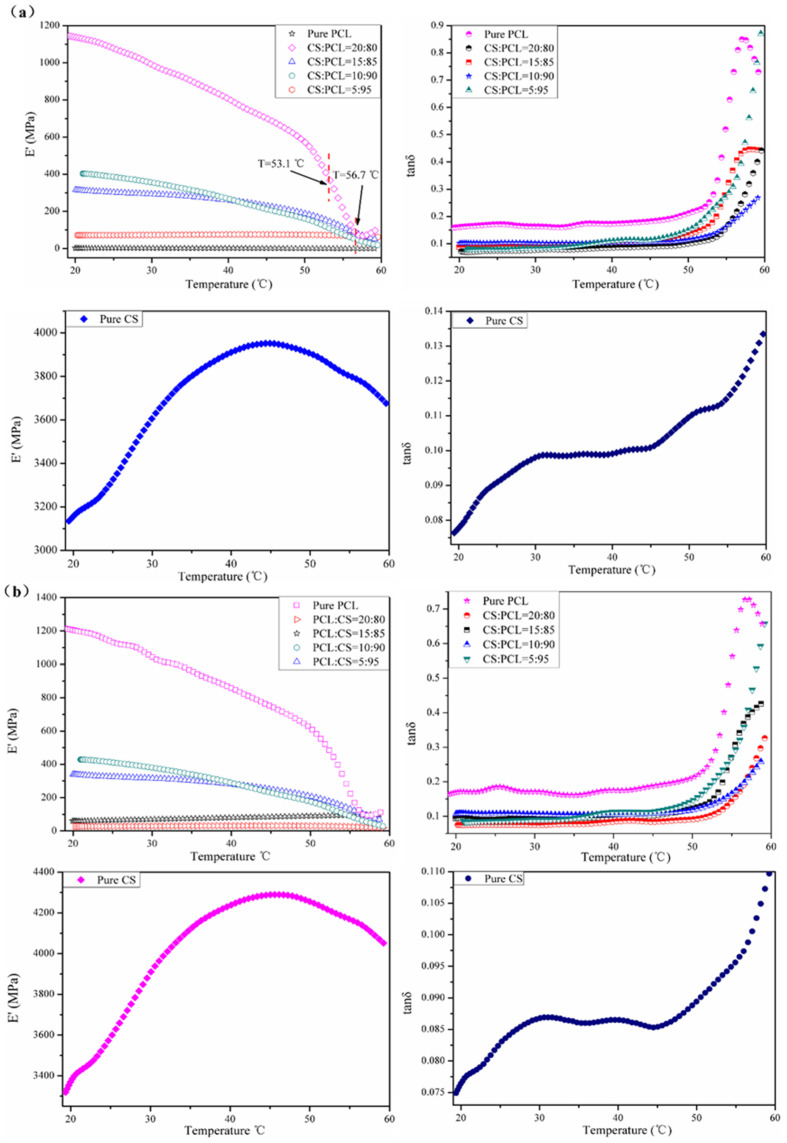
Temperature dependence of the storage modulus (E′) and loss factor (tan δ) of the tested membranes at constant frequency: 1 Hz (**a**) and 5 Hz (**b**), respectively.

**Figure 3 materials-14-05538-f003:**
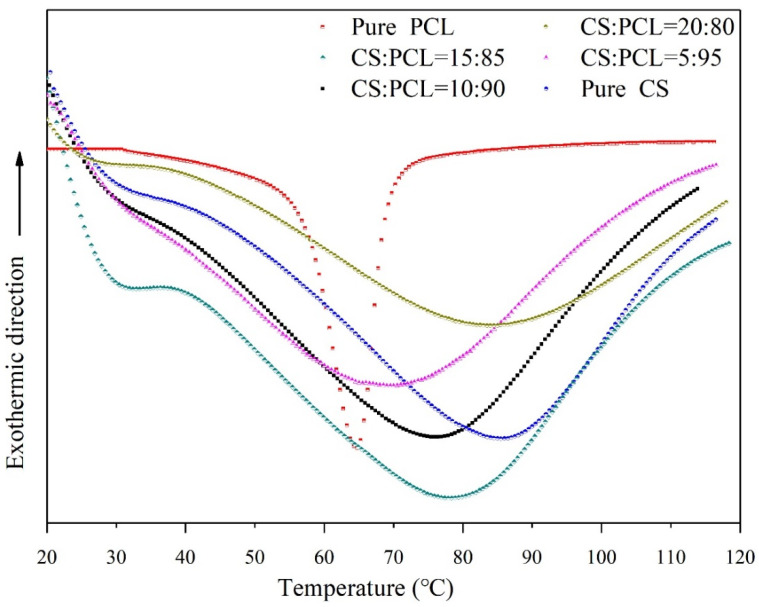
DSC thermograms of the tested membranes.

**Figure 4 materials-14-05538-f004:**
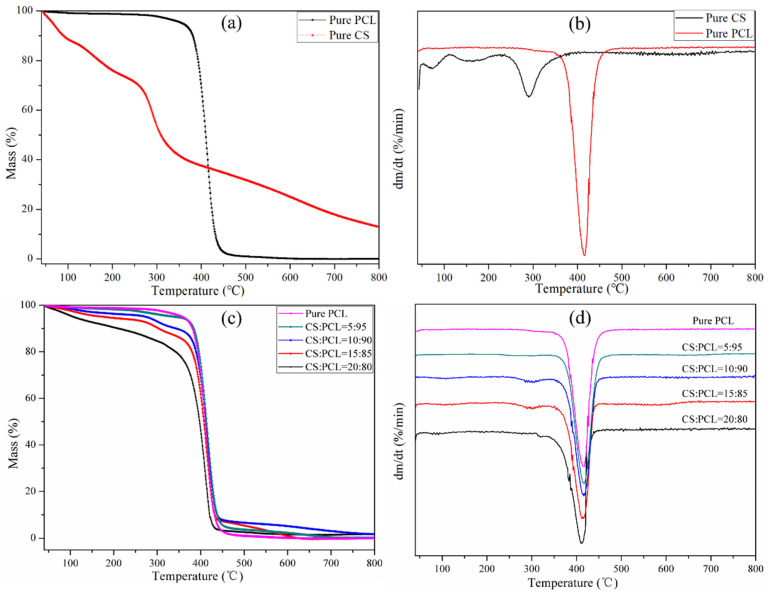
TG (**a**,**c**) and DTG (**b**,**d**) thermograms of the tested membranes.

**Table 1 materials-14-05538-t001:** Mechanical properties of the tested membranes with different blend ratios. Data are presented as mean ± SD.

CS:PCL	Tensile Strength (MPa)	Maximum Elongation (%)	Elongation at Break (%)
0:100	20.981 ± 0.447	210.74 ± 0.96	210.08 ± 0.53
5:95	20.289 ± 0.198	201.26 ± 0.78	205.82 ± 0.28
10:90	20.036 ± 0.283	196.27 ± 0.49	198.72 ± 0.62
15:85	17.764 ± 0.475	153.88 ± 0.74	162.88 ± 1.03
20:80	16.691 ± 0.231	96.61 ± 0.56	101.41 ± 0.77
100:0	12.363 ± 0.536	13.60 ± 0.73	19.29 ± 0.46

**Table 2 materials-14-05538-t002:** The mean porosity and the specific pore features of the tested membranes.

CS:PCL	Mean Porosity (%)	Specific Surface Area		Mean Pore Volume		Mean Pore Size	
		Cumulative Adsorption (m^2^/g)	Cumulative Desorption (m^2^/g)	Cumulative Adsorption (cm^3^/g)	Cumulative Desorption (cm^3^/g)	Adsorption (μm)	Desorption (μm)
0:100	96.74	8.6624	10.6078	0.59773	0.68114	0.0964	0.1093
5:95	89.18	4.6281	4.8766	0.19456	0.21097	0.0217	0.0278
10:90	85.61	4.2417	4.5961	0.18399	0.19015	0.0199	0.0259
15:85	80.96	3.9348	3.9914	0.16932	0.18273	0.0186	0.0194
20:80	67.55	2.1565	1.9287	0.07765	0.09563	0.0079	0.0096
100:0	60.29	1.0467	1. 3726	0.01536	0.01756	0.0035	0.0047

**Table 3 materials-14-05538-t003:** Thermal properties of the tested membranes with different blend ratios.

CS:PCL	*T_m_* (°C)	Δ*H_m_* (J/g)
0:100	64.8	56.74
5:95	69.8	61.93
10:90	77.6	79.65
15:85	77.9	80.13
20:80	78.1	81.97
100:0	86.3	97.62

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
