# Peer review of "Thermal Stability and Dynamic Mechanical Properties of Poly(ε-caprolactone)/Chitosan Composite Membranes"

_materials, 2021, doi:10.3390/ma14195538_

Round 1

Reviewer 1 Report

It would be worth describing the materials used in more detail

Author Response

Attached is the reply to your comments.Thanks for your carefully checking, we will write and check the manuscript elaborately and prudentially.

Reviewer 2 Report

This paper deals with the influence of blend ratio of poly(ε-caprolactone) (PCL) and chitosan (CS) on mechanical properties, porosity, surface wettabillity, dynamic mechanical viscoelasticity and thermal stability of laboratory prepared composite membranes. In order to prepare the CS-PCL membranes chitosan flakes were added (0-20%, w/w; 5% increase step) to dissolved PCL with subsequant liophylization of obtained solutions. 100% PCL and 100 % CS materials were also prepared and analysed.

The manuscript is well prepared and well written. However, there are few details that need to be adressed before it is to be published. As chitosan is a polymer of variable length, or to be more precise it is a mix of compounds and not an individual component, a GPC/SEC analysis should have been performed in order to determine its 'exact' molecular weight. This is due to the fact that some of the obtained results would have been easier to explain (e.g. melt temperatures) if the exact MW of both starting compounds is known. Also, the results of mechanical properties and those for porosity lack statistical analysis, which should be added. In subsection 3.5.Thermal properties and thermal stability, the Authors use the Tg marking for transition temperature and ΔHg for enthalpy. As obtained values (given in Table 3) are clearly related to melt and not to glass transition temperature (enthalpy) the markings should be altered to Tm and ΔHm. In DSC calorimetry, subscript g usually denotes that the given value is related to the glass transition temperature which for PCL is around -60 °C and for (unpurified) CS around 203 °C. 

As for more specific remarks, there are few language errors (e.g. hydrophily in line 15), and in some cases references should be added. More preciselly the sentences 'However, due to the difficult formation of intramolecular…' (lines 40-43) and 'Moreover, due to the repeat units of non-polar methylene and polar ester groups…' (lines 49-53) should be accompanied with references.

Author Response

(The authors gave the same response as above.)

Reviewer 3 Report

This work presents "Thermal stability and dynamic mechanical properties of the poly(ε-caprolactone)/chitosan composite membranes". The aim was improving properties of PCL by blending it with chitosan additive and fabricate a biocompatible composite.  Developing biocompatible composites is an important issue for academic. Although the presented results can be interesting for some readers, however, the manuscript needs to be revised to address the below mentioned comments. The existed critical drawbacks, inadequate results, and deficiencies make it very incomprehensible for publication.

1- The authors should clearly explain the innovation and importance of their work on the introduction of the manuscript. They should justify the value of the work and compare their work with previously similar published papers. The introduction section needs to be elaborated.

2- In Abstract, line 11, please change (PCL:CS, w/w) to (CS:PCL…).

3- Please provide chemical structure of PCL and CS and propose their interaction in the revised version of manuscript.

4- The manuscript does not present any characterizations of the prepared papers such as FTIR and SEM.

5- Please include a picture of different composites.

6- The previous work by the authors (Ref. 27) is not discoverable by search. Please provide the DOI or a PDF of the published work.

7- For tensile strength measurement: please write size of samples and cross-head speed.

8- Because in this study CS is an additive and blends with PCL in with different addition amount from 5 to 20 %, it is recommended that the authors change CS-PCL composite with PCL-CS composite.

9- In line 117-118, “With increasing PCL content in the composite ….. properties of CS-PCL membrane enhanced significantly.” Since in this study CS is an additive, this conclusion is not correct. In fact, with increasing CS content in the composite…. Properties of PCL-CS membrane diminished. This sentence and other similar sentences and conclusion in the manuscript must be revised.

10 -Line 135: Please check the ratio. The cumulative desorption of 0.19456 cm3/g is related to CS:PCL 5:95.

11- Line 140: “Redundant increase in PCL content…” similar with comment 9.

12- This manuscript is written like a report and not like a scientific paper. The experimental data are presented descriptively, without too many critical comments or comparisons with other studies in the literature.

Author Response

(The authors gave the same response as above.)

Round 2

Reviewer 2 Report

The Authors have clearly rewised their manuscript, added some data missing in the original version of the manuscript, and properly adressed certain details according with the reviewers' comments. However, there is still one major detail that have not been adressed, and that is the statistical analysis of data given in Tables 1 and 2. The Authors did explain that the statistical analysis was done, but there is no data on it. Therefore, prior to publication, this should (must) be adressed. Simple affixing of superscripts to means that are (not) significantly different from each other would suffice.

Author Response

Reply: we have added supplementary analysis of data given in Tables 1 and 2 and made the result comparisons with relevant references in section 3.1 and 3.2.

Reviewer 3 Report

The manuscript has been improved and can be published. Thanks to the authors for corrections.

Author Response

Thank you for your kind review and encouragement.